# Dynamic Simulation Analysis of the Working Process of the Picking Mechanism of a Sugarcane Leaf Cutting and Returning Machine

Zilong Ye, Yuxing Wang *, Yanqin Tang, Zhiguo Qiu, Wenhui Luo, Guofeng Ren and Qingxu Zhao

College of Engineering, South China Agricultural University, Guangzhou 510642, China
* Correspondence: scauwyx@scau.edu.cn

**Abstract:** Leaf–device interaction can effectively be modeled with a finite element model when proper finite element model parameters are applied. In order to investigate the contact mechanism of picking up sugarcane leaf during the operation of a sugarcane leaf cutting and returning machine, a geometric solid model of sugarcane leaf picking was established. A finite element numerical model to analyze the large deformation problem of flexible bodies was developed in LS-DYNA to simulate the picking process of the returning machine. A dynamic simulation of the sugarcane leaf-picking process was carried out to obtain the change of stress field and the motion posture of the sugarcane leaf and the elastic teeth. The picking process of the picking mechanism, the change in posture of the sugarcane leaves, the change in stress on the sugarcane leaf, the change in the bending angle of the sugarcane leaf and the change in stress on the elastic teeth were analyzed in detail. The results showed that the picking process can be divided into four stages: picking, lifting, pushing and retrieving. The posture changes of sugarcane leaf are "C", logarithmic curve, wavy shape and "V", in turn. During the picking process, the sugarcane blade showed some breakage, the sugarcane vein remained intact, and the elastic teeth did not fail. During the whole picking cycle, the maximum Von Mises stress of the blade, vein and elastic teeth were 22.8 MPa, 17.5 MPa and 900 MPa, respectively. An evaluation criterion of bending angle was creatively put forward to measure the bending deformation of leaves. The trend in the sugarcane leaf bending angle shows that it is largely variable, gradually decreasing, fluctuating and increasing with interval fluctuations. The working process of the picking mechanism was observed through a quick camera experiment. Comparing the experiment with the simulation, the changing trend of the simulation data and experimental data was generally similar. The experimental and simulation values of the average sugarcane leaf bending angle were 27° and 19°, respectively. The relative error of the average bending angle was 29.6%. It was concluded that the developed finite element model is substantial and could be applied to optimize and improve the picking mechanism. In addition, some references were provided for the contact mechanism between the picking mechanism and the sugarcane leaf.

**Keywords:** sugarcane leaf; returning machine; picking mechanism; finite element; dynamic simulation



## 1. Introduction

China's sugarcane-planted area in 2020 is 1353.38 thousand hectares, accounting for more than 85% of the sugar-planted area [1]. However, the degree of mechanized harvesting of sugarcane in China is very low, with the level of mechanization of sugarcane harvesting in China being around 3.2% in the 2019/2020 crushing season [2]. After the harvest of sugarcane, a large number of sugarcane leaves fall off, and the germination of sugarcane root and field management are seriously affected. Therefore, it is necessary to treat the sugarcane leaves left in sugarcane fields. In countries where the sugar cane industry is more developed, such as Brazil, India and Australia, large cutter combine harvesters with collection trucks are used to harvest the cane and shred the leaves, or tractors with disc

harrows are used to crush the leaves [3]. Domestic scholars have conducted a lot of research on the technology of returning cane leaves to the field, and have developed a series of sugar cane leaf crushing and returning machines [4–6]. However, these return machines have problems with their own weight and energy consumption, which can cause serious soil compaction and damage to the perennial roots. Therefore, South China Agricultural University has developed a light electric sugarcane leaf cutting and returning machine [7,8] which can be remotely controlled and does not damage perennial roots. At present, the third generation electric sugarcane leaf cutting machine has been developed. However, the picking contact mechanism of electric sugarcane leaf cutting and returning machine is still unclear, and there are large gaps in the research on the interaction between the sugarcane leaf and the picking mechanism. Therefore, it is necessary to analyze the picking process of sugarcane leaves in detail.

The contact between the picking mechanism and the sugarcane leaf faces the problem of large deformation of the flexible bodies. The common methods for analyzing the interaction between two flexible bodies are the finite element method and the discrete element method. The discrete element method has many problems; there are too many input parameters, the parameters need to be calibrated, and the calculation is huge. In contrast, the finite element method has a small calculation scale and is easy to accomplish. ANSYS/LS-DYNA software is commonly used to analyze non-linear dynamics problems and can describe the stress distribution of flexible bodies under large deformation conditions more accurately, and the simulation can save a lot of cost and time in field experiments. Cui et al. [9] modelled the maize stalk in ANSYS and investigated the process of the harvesting mechanism on the stalk. Dai et al. [8] carried out a simulation of the working process of a tracked flexible knife cutting and returning machine with the aid of LS-DYNA, and simulated the cutting of cane leaves by a high-speed rotating nylon flexible knife. Peng et al. [10] used LS-DYNA to establish a finite element model of river ice collision on a pier. The finite element model for calculating the collision force between river ice and a flexible pier was also studied. Park et al. [11] investigated the high-speed impact energy absorption properties of pure and STF-impregnated Kevlar fabrics using LS-DYNA. Xu et al. [12] used a finite element method to simulate the dynamic characteristics of large deformations during the unfolding of a flexible spinning web. Xie et al. [13] constructed a finite element model of the sugarcane defoliation process using hypermesh and LS-DYNA software. Zhu et al. [14] used the explicit analysis method of ANSYS software to simulate the ballistic performance of SiC-UHMWPE flexible laminated structures. Zhang et al. [15] conducted a dynamic cutting mechanics simulation based on LS-DYNA for a double-layer flexible model of cereal stalks, and carried out validation experiments using a home-made cutting experiment rig. Chen et al. [16] established a finite element simulation model for cutting roots of garlic, and explored the effect of tool parameters on cutting forces. Sreenivas et al. [17] used LS-DYNA to simulate automotive panels made of twill-woven Kenaf and plain-woven Kevlar flexible polymer composites to study low-velocity impact behavior. Lam H.W.K. et al. [18] used LS-DYNA and NIDA-MNN for numerical simulation to analyze the dynamic response of flexible barriers with different structural forms. Dakshitha Weerasinghe et al. [19] used LS-DYNA to investigate for the first time the effect of fatigue loading on the yarn pull-out force of flexible coated Twaron fabrics, and developed an effective mesoscale numerical modelling approach to simulate ballistic impacts.

It is rare to take flexible elastic teeth as the research object. This paper precisely used LS-DYNA software to establish a finite element model of the picking system, to analyze the process of picking sugarcane leaves using the picking mechanism, and to verify the rationality of the finite element model through experiments. The characteristics of the relationship between the picking mechanism and the cane leaves were revealed, and this paper also provided some theoretical basis for the improvement and design of future picking mechanisms. Furthermore, the evaluation criterion of bending angle was innovatively put forward to measure the bending deformation of the leaf.

## 2. Materials and Methods

### 2.1. Working Mechanism of the Returning Machine

The working mechanism of the electric sugarcane leaf cutting and returning machine is shown in Figure 1, and is mainly composed of the elastic teeth roller picking mechanism, the conveying brush roller and the cutting knife roller. The working mechanism is driven by two working motors, one of which transmits power to the cutting tool roller through a belt drive, while the other transmits power to the picking mechanism and the conveying brush roller through a chain drive. When the returning machine is in operation, the elastic teeth of the picking mechanism pick up the sugarcane leaves on the sugarcane field, and the leaves are fed along the guard of the picking mechanism towards the underside of the brush roller. The speed of the brush roll is greater than the speed of the picking mechanism. Additionally, the brush bristles will separate the sugarcane leaves from the elastic teeth, sweep and push them to the working area of the knife roller. The knife roller cuts off sugarcane leaves with the cooperation of moving and fixed knives, and scatters them back to the field.

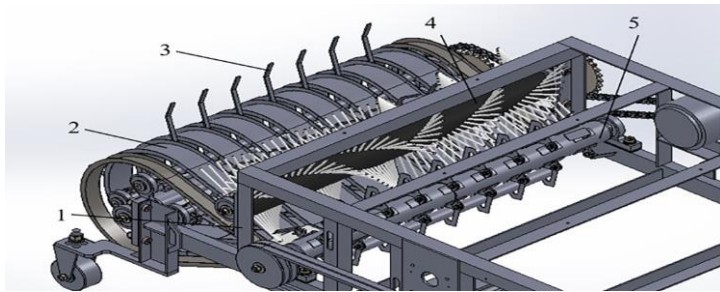

**Figure 1.** The working mechanism of the returning machine (1: door frame; 2: guard plate; 3: elastic teeth roller picking mechanism; 4: conveying brush roller; 5: cutting knife roller).

### 2.2. Three-Dimensional and Finite Element Modelling

#### 2.2.1. Three-Dimensional Model of the Picking System

The picking system mainly includes two models: the picking mechanism model and the sugarcane leaf model. Whether the model is reasonable or not directly affects the accuracy and reliability of the simulation results. Because of the complex shape of the picking mechanism, the professional three-dimensional software SolidWorks is used to create its three-dimensional geometric model. The structural parameters of the picking mechanism are shown in Table 1. The schematic diagram of the picking system structure is shown in Figure 2.

**Table 1.** Structural parameters of the picking mechanism.

| Parameters | Value |
| --- | --- |
| Radius of base circle (mm) | 41 |
| Radius of roller plate (mm) | 60 |
| Length of crank (mm) | 40 |
| Length of elastic tooth (mm) | 150 |
| Angle between elastic teeth and crank (°) | 52 |
| Three arc radius (mm) | 41/53/89 |
| Angle of upper guard plate (°) | 24 |
| Number of elastic tooth rod | 4 |

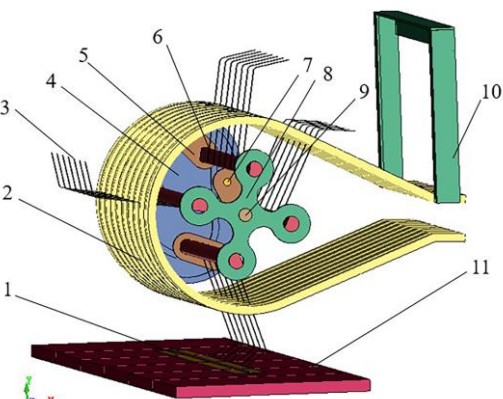

**Figure 2.** Schematic diagram of the pick-up system structure (1: sugarcane leaf; 2: guard plate; 3: elastic teeth; 4: cam plate; 5: crank linkage; 6: elastic teeth rod; 7: roller wheel; 8: roller plate; 9: power shaft; 10: door frame; 11: ground).

The picking mechanism is the key component of the sugarcane leaf cutting and returning machine, and is mainly composed of the guard plate, elastic teeth, cam plate, crank linkage, elastic teeth rod, roller wheel, roller plate, power shaft and so on. The geometric parameters of the elastic tooth are shown in Figure 3.

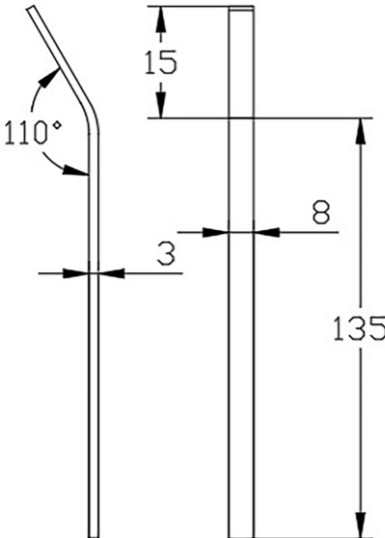

**Figure 3.** Geometric parameters of the elastic tooth.

The roller plate is fixed on both sides of the power shaft, and the tooth rod is hinged to the roller plate. The elastic tooth and the crank linkage are simultaneously fixed on the elastic tooth rod. The other end of the crank linkage is provided with a roller wheel which rolls in the cam slide of the cam plate. When the picking mechanism is in operation, the power shaft drives the roller plate to rotate at a constant speed, causing the circumferential tooth rod to move in a circular motion. Because of the restriction of the cam slide on the roller, the crank linkage drives the roller wheel to move according to its prescribed track. The tooth rod is hinged to the roller plate. The movement of the teeth is therefore controlled both by the circular rotational movement of the tooth rod and by the movement of the roller wheels in the cam slide, which rotate around the power shaft and swing around the elastic tooth rod.

Mature sugarcane leaves are approximately 80–150 cm in length and 2–5.39 cm in width [20]. Take the sugarcane leaf model with a length of 100 cm and a width of 4 cm; the sugarcane leaf is made up of two parts, the vein and the blade. The crescent shape of the leaf

vein cross-section is simplified to a semicircle with a radius taken as 0.24 cm, and the leaf blade cross-section is simplified to a rectangle with a thickness taken as 0.3 mm [21]. When establishing the geometric model of the pickup mechanism, in order to reduce the model scale and shorten the calculation time, it is necessary to simplify the model appropriately by ignoring the rounded corners and chamfers of the cam plate, roller plate and bolt holes of guard plate, and so on.

### 2.2.2. Finite Element Model of the Picking System

The finite element model was built using the display dynamics software ANSYS/LS-DYNA, which does not have a fixed unit system and has to be defined by the user. In the numerical simulation and analysis of the pick-up system, the kg-m-s unit is used, and all other units can be derived on the basis of this unit system. The tetrahedral meshing method is simple, adaptable to the model and suitable for all kinds of complex geometric models. Hexahedral meshes have greater advantages in terms of computational accuracy, mesh quality and speed of computational convergence for simple models. For parts that are much thinner in one direction compared to the other, shell elements are generally used to divide the mesh. Therefore, tetrahedral grids are used to mesh the guard plate, cam plate, power shaft, crank linkage, elastic tooth rod, roller plate, roller wheel and door frame. Hexahedral grids are used to mesh sugarcane leaf . A shell element is used to mesh the elastic teeth. The total number of cells for the entire finite element model is 546,835, and the total number of nodes is 327,439.

The elastic–plastic isotropic model (* MAT _ Plastic _ Kinetic) is selected as the sugarcane leaf model. This model is suitable for simulating isotropic and kinematic hardening plasticity, and the rate effect is considered. The hardening model used in this paper is the Prager model. The multi-segment linear elastic–plastic model (* Mat _ Piecewise _ Linear _ Plasticity) is selected as the elastic model. This model can describe the constitutive relation of materials by defining the effective stress–strain curve considering the strain rate effect.

To define the effective stress–strain curve, it is necessary to obtain the engineering stress–strain curve of the material. The engineering stress–strain curve for 65 Mn spring steel is referenced in this paper [22]. The engineering stresses and strains are converted into real stresses and strains by means of Equations (1) and (2).

$$\varepsilon_t = ln(1 + \varepsilon_e) \tag{1}$$

$$\sigma_t = \sigma_e(1 + \varepsilon_e) \tag{2}$$

$$\varepsilon_p = \varepsilon_t - \sigma_t/E \tag{3}$$

In Equations (1)–(3), $\varepsilon_t$ represents real strain, and $\varepsilon_e$ represents engineering strain. $\sigma_t$ represents the true stress (Pa); $\varepsilon_e$ represents the engineering stress (Pa); $\varepsilon_p$ represents the effective strain; and $E$ represents the modulus of elasticity (Pa).

Equation (3) is then used to calculate the effective strain from the true stress and strain. Taking the part of the effective strain greater than 0, as the maximum effective plastic strain is considered, the part of the unloading drop is wiped out and the curve is kept as a rising state. Finally, an effective stress–strain curve is obtained.

Based on the required material properties, the mechanical parameters of the cane leaves are referenced from experiments on the mechanical parameters of cane leaves carried out by Zheng et al. [23] at the Institute of Agricultural Machinery, Chinese Academy of Tropical Agricultural Sciences. The mechanical parameters of the elastic teeth are set according to the parameters of 65 Mn spring steel [24]. The material parameters for each component are shown in Table 2.

**Table 2.** Mechanical and material parameters of models.

| Models | Parameter | Parameter Values |
|---|---|---|
| Leaf blade/leaf vein | Elastic modulus (MPa) | data 1281/716 |
| | Shear modulus (MPa) | 493/272 |
| | Tensile strength (MPa) | 33/22 |
| | Shear strength (MPa) | 5.7/7.8 |
| | Poisson's ration | 0.29/0.31 |
| | Hardening parameter | 0.0 |
| Elastic teeth | Density (kg.m$^{-3}$) | 7800 |
| | Elastic modulus (Pa) | $2.0 \times 10^{11}$ |
| | Poisson's ratio | 0.269 |
| Guard plate, cam plate, power shaft, crank linkage, elastic tooth rod, roller wheel, roller plate, door frame, ground | Density (kg.m$^{-3}$) | 7850 |
| | Elastic modulus (Pa) | $2.1 \times 10^{11}$ |
| | Poisson's ratio | 0.3 |

The finite element model of the picking system consists of two main models, the picking mechanism and the sugarcane leaf. The sugarcane leaves are parallel to the power shaft and placed on the ground surface. In the finite element model, the shell element is used for the elastic teeth, while the solid element is used for the guard plate, the cam plate, the power shaft, the crank linkage, the elastic tooth rod, the roller plate, the roller wheel, the door frame, the ground and the sugarcane leaf. The finite element model of the picking system is shown in Figure 4.

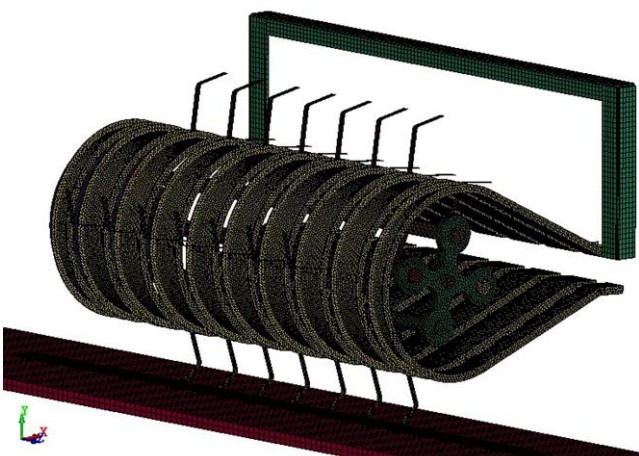

**Figure 4.** Finite element model of picking system.

The speed of roller plate was set at 35 r/min, and the axis of rotation was in the positive direction of the z-axis, rotating counterclockwise. The forward speed of cam plate, power shaft, roller plate, guard plate and door frame was set to 0.4 m/s, and the forward direction was the negative direction of the x-axis. The guard, cam plate, crank linkage, elastic tooth rod, roller wheel, roller plate, power shaft, door frame and ground were set as rigid bodies. The sugarcane leaf and elastic teeth were set as flexible bodies. The acceleration of gravity in the y direction was applied to the sugarcane leaf and elastic teeth. The contact relationship between the sugarcane leaves is single-surface contact, and the contact relationship between sugarcane leaves and the other parts is surface-to-surface contact. The penalty function is used to deal with the contact problem. The model includes 10 groups of contacts. The specific keywords and friction coefficients are shown in Table 3. The termination time of the simulation is set to the time of one cycle of elastic teeth movement, which is 1.8 s. Once the rest of the keywords have been set, the K file can be solved for.

**Table 3.** Contact settings.

| Contact Model | Keywords | Static Friction Coefficient | Dynamic Friction Coefficient |
|---|---|---|---|
| Sugarcane leaf and elastic teeth | AUTOMATIC_SURFACE_TO_SURFACE | 0.6 | 0.58 |
| Sugarcane leaf and ground | AUTOMATIC_SURFACE_TO_SURFACE | 0.1 | 0.05 |
| Sugarcane leaf and guard plate | AUTOMATIC_SURFACE_TO_SURFACE | 0.1 | 0.05 |
| Sugarcane leaf and door frame | AUTOMATIC_SURFACE_TO_SURFACE | 0.1 | 0.05 |
| Sugarcane leaf and sugarcane leaf | AUTOMATIC_SIN-GLE _SURFACE | 0.1 | 0.05 |
| Roller plate and power shaft | AUTOMATIC_SURFACE_TO_SURFACE | 0.1 | 0.05 |
| Roller plate and elastic teeth rod | AUTOMATIC_SURFACE_TO_SURFACE | 0.1 | 0.05 |
| Crank linkage and roller wheel | AUTOMATIC_SURFACE_TO_SURFACE | 0.1 | 0.05 |
| Cam plate and roller wheel | AUTOMATIC_SURFACE_TO_SURFACE | 0.1 | 0.05 |
| Cam plate and power shaft | AUTOMATIC_SURFACE_TO_SURFACE | 0.1 | 0.05 |

## 3. Simulation and Analysis of the Sugarcane Leaf-Picking Process

### 3.1. Analysis of the Working Process of the Pick-up Mechanism

The elastic teeth in the elastic teeth roller picking mechanism make up a complex composite of motion elements. While following the advance of the sugarcane leaf cutting and returning machine and the rotation of the roller plate, it oscillates regularly and repeatedly in relation to the roller plate. The rotation–position relationship of the roller plate corresponding to the posture change of the elastic teeth in the movement process is defined as the pick-up phase of the elastic teeth, which is expressed by the rotation angle of the roller plate. After observation, it is found that the roller plate rotates clockwise for one cycle, and each elastic tooth continuously completes a pick-up cycle of four stages: "pick-up", "lift-up", "push-up" and "empty return". The phase diagram of the picking mechanism is shown in Figure 5.

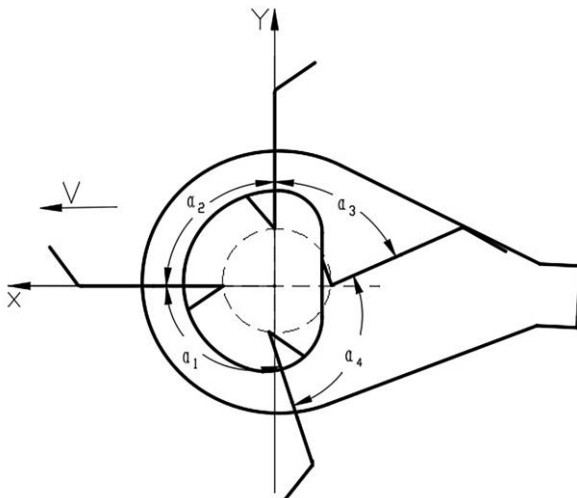

**Figure 5.** Phase diagram of the picking mechanism.

(1) The pick-up stage (elastic tooth phase angle $\alpha_1$) imitates the situation in which people pick up sugarcane leaves with a plow. The elastic teeth are initially inserted between the ground and the sugarcane leaves at a certain inclination. The ends of the teeth then start to contact the sugarcane leaf, and the teeth gradually move to the horizontal position. Under the combined absolute speed of the roller plate rotation and the horizontal advancement of the returning machine, the end speed of the elastic teeth begins to rise gradually, so that the sugarcane leaf can be picked up and rotated clockwise along with the elastic teeth.

(2) The lift-up stage (elastic tooth phase angle $\alpha_2$): At the beginning of the lift-up stage, the elastic teeth are in the same radial direction as the roller plate and in a horizontal state. As the roller plate rotates, the linear speed at the end of the elastic teeth changes from

vertical upwards to rear upwards in order to lift and transport the sugarcane leaf above the guard plate.

(3) Push-up stage (elastic tooth phase angle $\alpha_3$): At the beginning of the push-up phase, the teeth are basically in a plumb position, and the relative velocity direction is horizontal backwards, pushing the sugarcane leaf backwards.

(4) Empty return stage (elastic tooth phase angle $\alpha_4$): At the beginning of the empty return stage, the elastic teeth have shrunk into the guard plate. With the continuous rotation of the roller plate, the elastic teeth that carry out relative movement will return to the initial position of the pick-up stage after finishing the pick-up, lift-up and push-up stages, so as to prepare for the next picking cycle. During this time, the elastic teeth are not in contact with the sugarcane leaf and the ground, but their angular acceleration of oscillation relative to the roller plate is high, in order to return to the original picking position as soon as possible at the start of the next picking phase.

### 3.2. Analysis of Sugarcane Leaf Posture Picking Process

The sugarcane leaf stress cloud and the sugarcane leaf motion posture are shown in Figure 6.

When T = 0 s, before the picking mechanism has started working, the picking teeth do not touch the sugarcane leaf, which was placed horizontally on the ground.

When T = 0.25 s and the teeth were in the pick-up stage, the sugarcane leaf was picked up by the force of the teeth. The middle part of the sugarcane leaf in contact with the teeth gradually became convex and moved upwards. The two sides of sugarcane leaf that are not in contact with the elastic teeth gradually droop. The overall shape of the sugarcane leaf was approximated by the letter "C", rotated at 90°.

When T = 0.41 s, the middle part of the sugarcane leaf moved upwards and hit the guard; the leaf was then subjected to the force of the guard on it, the frictional force of the teeth on it and its own gravity. The sugarcane leaf started to slide down the sloping surface of the teeth, and hung on to the curved, hooked part of the teeth.

When T = 0.85 s and the elastic teeth were in the lift-up stage, the movement was smoother and the sugarcane leaf was lifted above the guard as the elastic teeth rotated. The sugarcane leaf slid from the curved, hooked part of the elastic teeth along the surface of the elastic teeth towards the guard plate, and the two sides of the sugarcane leaf touched the guard plate before the middle part. The overall shape of sugarcane leaves is, approximately, a logarithmic curve.

When T = 1.47 s and the elastic teeth were in the push-up stage, the sugarcane leaf was subjected to the thrust of the elastic teeth on it, the frictional force of the guard on it and its own gravity at this stage. The sugarcane leaf was pushed backwards and downwards along the guard by the elastic teeth. The overall posture of the sugarcane leaf was approximated by a wavy shape.

When T = 1.80 s and the elastic teeth were in the empty return stage, the sugarcane leaf was subjected to the frictional force of the guard and its own gravity. The sugarcane leaf slid along the guard plate and collided with the door frame, then bounced back some distance. The overall posture of the sugarcane leaf was approximately in the shape of the letter "V".

The curve of the maximum Von Mises stress on the blade and vein is shown in Figure 7. Combined with Figure 6, the change of the equivalent force field of the sugarcane leaves is analyzed.

The sugarcane leaf consists of the blade and the vein. During the picking process, stress concentrations were mostly found in the vein and where it are connected to the leaf, where the sugarcane leaves are in contact with the elastic teeth, and where the sugarcane leaf collides with the guard plate. The stress concentration at the position where the sugarcane leaf is picked up and bent by elastic teeth was obvious.

Before contact with the door frame, the maximum stress occurred on either side of the point where the sugarcane leaf came into contact with the teeth or with the guard. After the contact between the sugarcane leaf and the door frame occurs at t = 1.53 s, the maximum value of the stress on the leaf blade occurred in the middle of the contact position between

the leaf and the two large guard plates close to the roller plate. The situation in which the maximum Von Mises stress of the blade appeared is shown in Figure 8. At this time, the blade was subjected to the downward pressure of the elastic teeth, the supporting force of the guard plates, the force of the vein, and gravity. The blade located between the guard plates has the largest deformation, so the Von Mises stress of the blade reached the maximum.

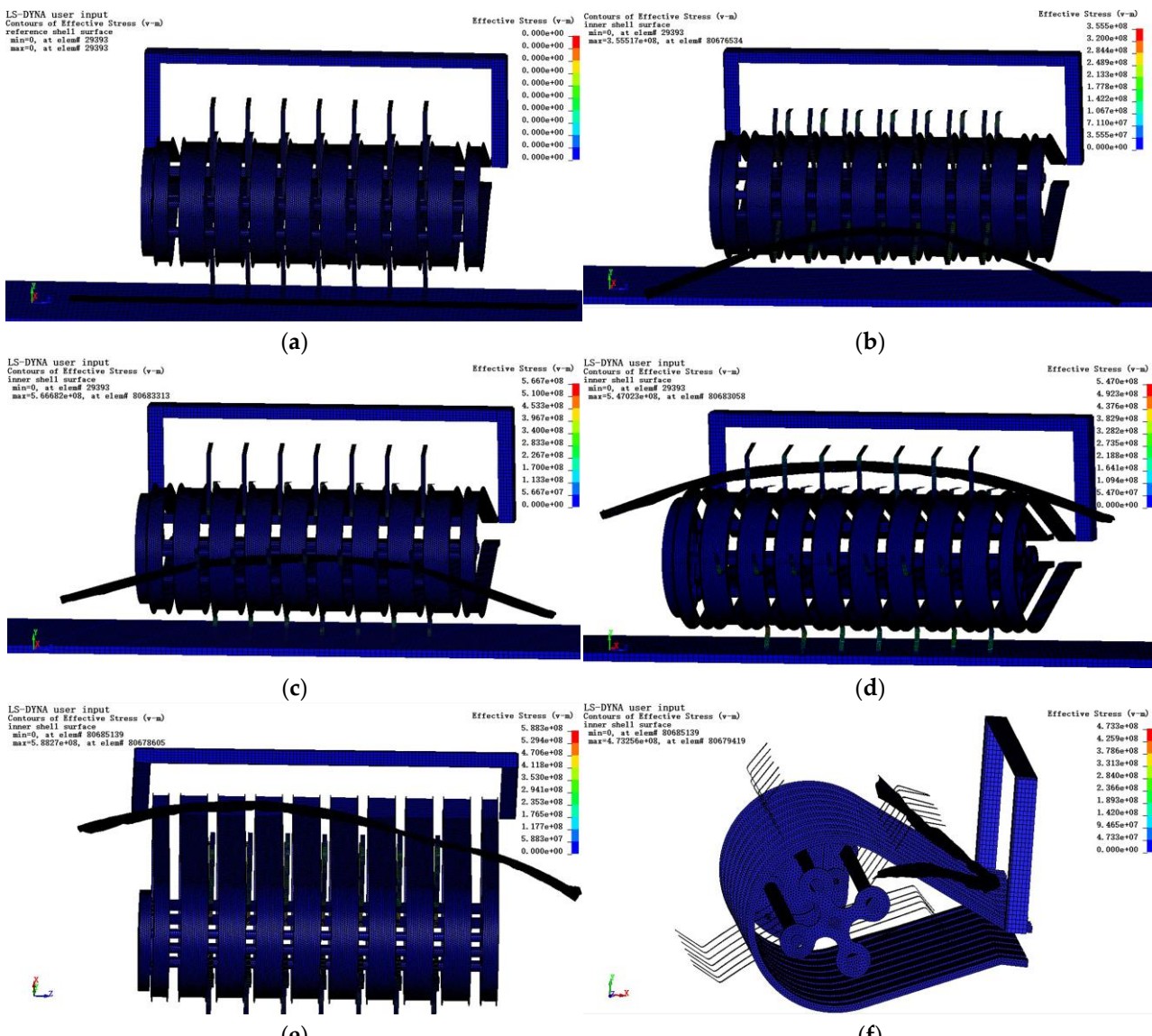

**Figure 6.** Stress clouds and motion posture of sugarcane leaf. (**a**) represents the stress and posture diagram of the cane leaf at T = 0 s; (**b**) represents the stress and posture diagram of the cane leaf at T = 0.25 s; (**c**) represents the stress and posture diagram of the cane leaf at T = 0.41 s; (**d**) represents the stress and posture diagram of the cane leaf at T = 0.85; (**e**) represents the stress and posture diagram of the cane leaf at T = 1.47 s; and (**f**) represents the stress and posture diagram of the cane leaf at T = 1.80 s.

At t = 1.55 s, the maximum value of stress on the leaf vein also occurred near the above position. At that time, the vein was subjected to the downward pressure of the elastic teeth, the supporting force of the guard plates, the force of the blade, and gravity. The deformation of the vein located between the guard plates is the largest, so the Von Mises stress of the vein there reaches the maximum. During the whole picking cycle, the maximum Von Mises stress of the blade and vein was 22.8 MPa and 17.5 MPa, respectively.

Stress on the blade reached its yield limit, and a slight breakage occurred. The stresses on the vein had not reached their yield limit, and the vein pattern remained intact.

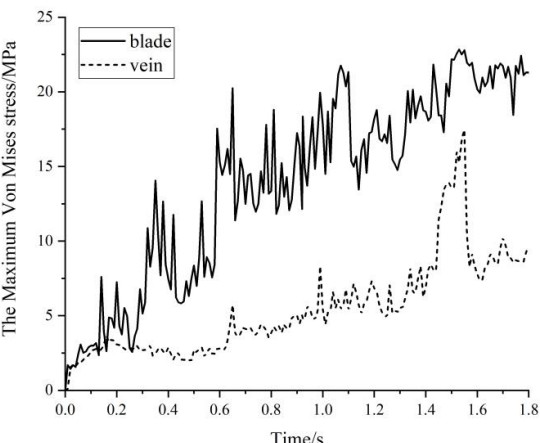

**Figure 7.** Curve of the maximum Von Mises stress of blade and vein.

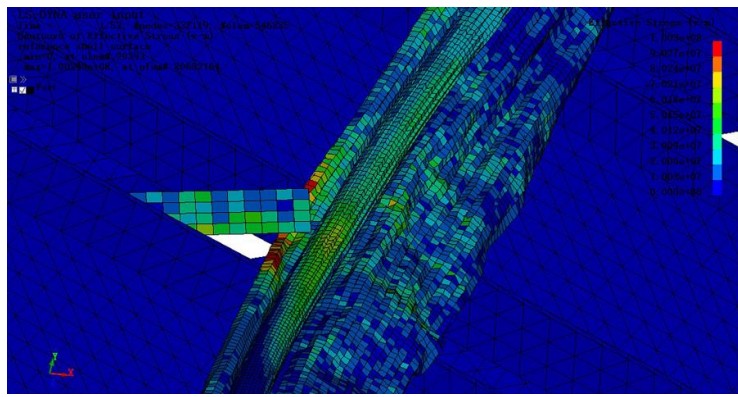

**Figure 8.** Situation in which the maximum Von Mises stress of the blade appears.

*3.3. Analysis of Stress Change of Elastic Teeth*

Figure 9 is the time curve of the maximum stress of the Von Mises of the elastic teeth in the picking process.

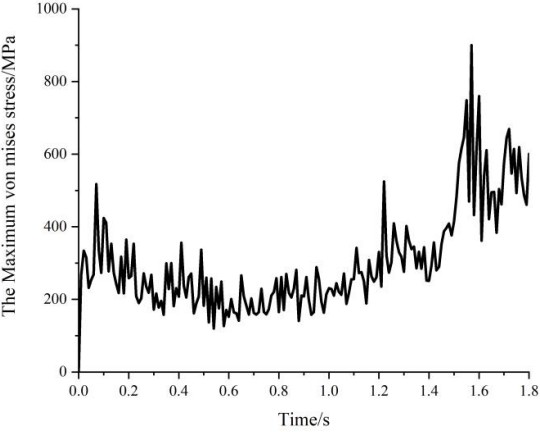

**Figure 9.** Maximum Von Mises stress–time curve of elastic teeth.

Figure 9 shows the effective stress distribution of elastic teeth at different times. From Figures 9–11, it can be seen that the effective stress on the teeth increased first during the picking process, reaching a global first peak of 518 MPa at t = 0.07 s, and then decreased.

Then, it fluctuated up and down in the range of 120~356 MPa and reached a second global peak of 525 MPa at t = 1.22 s. After that, it rose in a fluctuating way. Then, /it reached a maximum value at t = 1.57 s. The occurrence of maximum effective stress on the elastic teeth is shown in Figure 10.

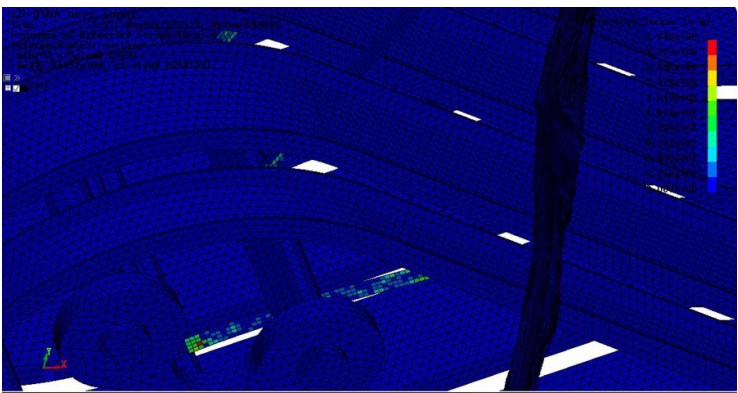

**Figure 10.** Occurrence of maximum effective stress on elastic teeth.

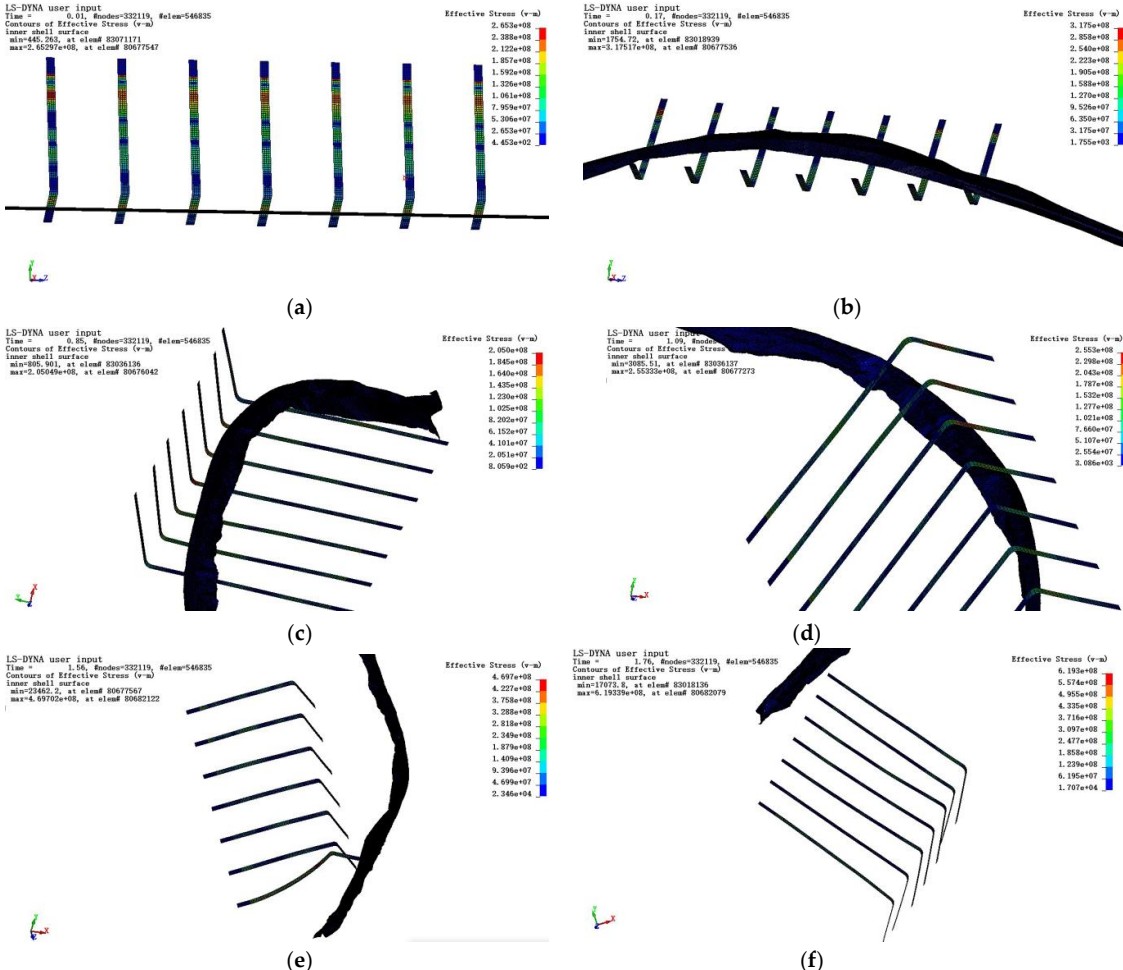

**Figure 11.** Effective stress distribution of elastic teeth at different times. (**a**) represents the effective stress distribution of the elastic teeth at T = 0.01 s; (**b**) represents the effective stress distribution of the elastic teeth at T = 0.17 s; (**c**) represents the effective stress distribution of the elastic teeth at T = 0.85 s; (**d**) represents the effective stress distribution of the elastic teeth at T = 1.09 s ;€ represents the effective stress distribution of the elastic teeth at T = 1.56 s; and (**f**) represents the effective stress distribution of the elastic teeth at T = 1.76 s.

At that time, the elastic teeth had just crossed the obstacle of the sugarcane leaf and were recycled into the guard plate. The elastic tooth swung around the elastic tooth rod to a certain extent. The maximum effective stress appeared at the root of the elastic tooth. Finally, the stress fluctuated in the range of 362~759 MPa. The yield limit and strength limit of 65 Mn spring steel are 821 MPa and 1183 MPa, respectively. The maximum Von Mises stress of the elastic teeth appeared at the root of the elastic teeth, and the maximum stress is 900 MPa. At this time, the stress value was greater than the yield limit and less than the strength limit, which indicated that the elastic tooth has plastic deformation at this time and that 65 Mn spring steel was in the strengthening stage. The elastic teeth will not fail in the process of picking up sugarcane leaf, and the multi-segment linear elastic–plastic model can be used to simulate the deformation of the elastic teeth.

As can be seen from Figure 11a, at the beginning of the picking motion, the inclined surface of the front end of the elastic teeth began to contact and cradle the sugarcane leaf, and the sugarcane leaf was elastically deformed. The sugarcane leaf exerted a reaction force on the elastic teeth, and the effective stress of the elastic teeth increased, among which the effective stress at the root of the elastic teeth increased more than at other parts of the elastic teeth.

As can be seen from Figure 11b, at t = 0~0.17 s, the elastic teeth were in the pick-up stage and the sugarcane leaf was gradually bent at both ends under the action of the elastic teeth. The sugarcane leaf exerted pressure on the elastic teeth, and there was an overall tendency for the equivalent stress on the elastic teeth to increase. The maximum stress occurred at the root of the elastic tooth on one side. The effective stress of the curved hook part of the elastic teeth increased obviously.

As can be seen from Figure 11c, at t = 0.17~0.85 s, the elastic teeth were in the lift-up stage and moved obliquely upward. The sugarcane leaf slid down along the inclined surface of the elastic teeth under the action of the elastic teeth. The effective stress of the elastic teeth showed an overall fluctuating trend, with the maximum stresses occurring in the curved hook part of elastic teeth.

As can be seen from Figure 11d, at t = 0.85~0.1.09 s, the elastic teeth were in the push-up stage and moved obliquely downward. The sugarcane leaf slid along the surface of the guard plate under the pressing action of the flat face at the tip of the elastic teeth. The effective stress of the elastic teeth fluctuated as a whole, and the maximum stress appeared at the flat face of the tip of the elastic teeth.

As can be seen from Figure 11e, at t = 1.56 s, the teeth were in the empty return stage and were retracted towards the guard plate. The sugarcane leaf slid down the guard plate towards the door frame at the end of the guard plate. Due to the large length of the sugarcane leaf and the bending of the sugarcane leaf on both sides, one of the side elastic teeth was prevented from entering the guard plate. The elastic teeth bent and pressed down the sugarcane leaf. The maximum stress occurred at the flat surface in the middle of the teeth.

As can be seen from Figure 11f, at t = 1.76 s, the elastic teeth were in the empty return stage. With the advance of the picking mechanism and the rotation of the roller plate, the elastic teeth had bounced over the sugarcane leaves and into the guard plate. The effective stresses on the elastic teeth showed an overall fluctuating trend, and the stresses on the various parts of the elastic teeth are relatively homogeneous in distribution.

### 3.4. Analysis of Changes in the Sugarcane Leaf Bending Angle

In order to observe the bending effect of the elastic teeth on both sides of the sugarcane leaf, points A and B on the sugarcane leaf were extracted in post-processing. Point A is the midpoint of the bend, wherein the sugarcane leaf contacts the elastic teeth, and point B is the midpoint of the end of the sugarcane leaf. The extraction point of the sugarcane leaf bending angle is shown in Figure 12.

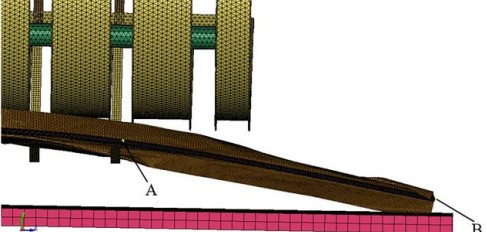

**Figure 12.** Extraction point of sugarcane leaf bending angle.

Then, the angle of bending of the sugarcane leaf on both sides (i.e., the angle between the bent section of the sugarcane leaf and the horizontal direction) was given by

$$\alpha = \arccos \frac{z_2 - z_1}{\sqrt{(x_2 - x_1)^2 + (y_2 - y_1)^2 + (z_2 - z_1)^2}} \tag{4}$$

In Equation (4), $\alpha$ represents the cane leaf bending angle (°). $x_1$, $y_1$, $z_1$ represent the coordinate values of point A. $x_2$, $y_2$, $z_2$ represent the coordinate values of point B.

Figure 13 represents the curve of the change in the bending angle of the sugarcane leaf during the picking process. From 0~0.14 s, the sugarcane leaf contacted the elastic teeth and the ground. The sugarcane leaf was subjected to the resultant force of the elastic teeth moving obliquely upward and the friction force of the ground. The bending angle of sugarcane leaf increased rapidly. The rate of change of the sugarcane leaf bending angle reached its maximum when the elastic teeth moved to 0.03 s. From 0.14~0.18 s, the middle part of sugarcane leaf was gradually bounced by the elastic teeth. The sugarcane leaf produced some rebound and the bending angle of the sugarcane leaf was reduced more slowly. From 0.18~0.22 s, the sugarcane leaf made contact with the underside of the guard and broke away from the ground at 0.19 s. The sugarcane leaf was subjected to the forces of the elastic teeth and the guard plate during this time, and the bending angle increased more slowly. From 0.22~0.58 s, the sugarcane leaf moved upwards with the rotation of the elastic teeth. Some rebound occurred in the sugarcane leaf on both sides. The overall trend in the bending angle of the sugarcane leaf showed that it was gradually becoming smaller. From 0.58~1.34 s, the sugarcane leaf was lifted by the elastic teeth to the uppermost part of the guard plate and then pushed downwards by the elastic teeth. The sugarcane leaf detached from the elastic teeth at t = 1.09 s, and then slid along the guard plate. At t = 1.26 s, some areas of the sugarcane leaf began to collide with the door frame. At t = 1.34 s, the bending angle of sugarcane leaf reached a maximum value of 27.75°. The overall trend in the folding angle of the sugarcane leaf during this period was gradually increasing. From 1.34~1.8 s, the sugarcane leaf slid down the guard plate to the end of the guard plate. The area wherein sugarcane leaf hit the door frame became larger.

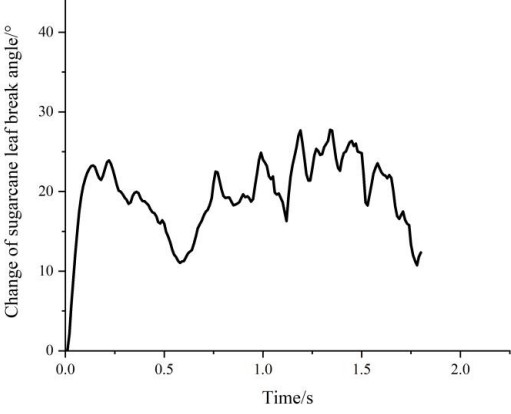

**Figure 13.** Change in the bending angle of the sugarcane leaf during the picking process.

The sugarcane leaves bounced some distance after the collision. The overall trend in the bending of the sugarcane leaf during this period is a gradual decrease. Finally, the bending angle changed to 12.34°.

### 3.5. Hourglass Energy–Time Curve

Typically, the simulation is valid at hourglass energies below 10% of the internal energy. The comparative curves of the changes in the internal energy of the model and the hourglass energy are shown in Figure 14.

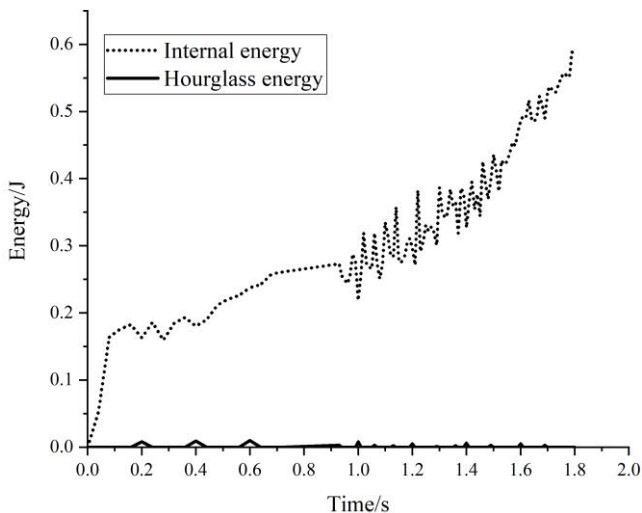

**Figure 14.** Comparative curves of changes in model internal energy and hourglass energy.

The ratio of hourglass energy to internal energy of the model is less than 10%, as shown in Figure 14, indicating that the hourglass energy set can effectively control the elementary distortion to achieve the desired simulation goal. Therefore, the grid division and parameter setting are correct. Additionally, the simulation results are reliable and can truly reflect the picking process of the sugarcane leaf.

## 4. Experiment and Analysis

### 4.1. Experimental Programme

In order to verify the finite element modelling and simulation results, the sugarcane leaf-picking process of the sugarcane leaf cutting and returning machine was analyzed using a quick camera method. The experimental principle is shown in Figure 15.

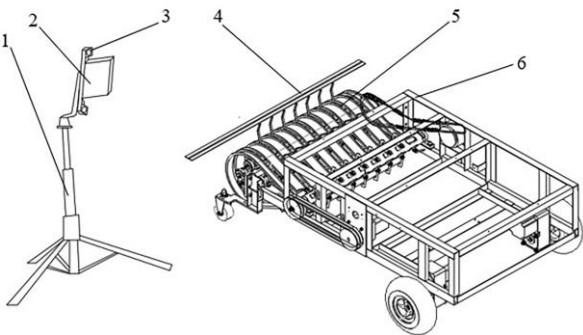

**Figure 15.** Schematic diagram of quick camera experiment (1: triangular support frame; 2: quick camera; 3: LED filler light; 4: sugarcane leaf; 5: picking mechanism; 6: sugarcane leaf cutting and returning machine).

Before the returning machine is started, a 48 megapixel Redmi K40 quick camera was used to ensure that the complete picking of the sugarcane leaf was recorded. The rotational

speed control of the roller plate and the forward speed control of the machine were the same as in the simulation model. The recording rate was set to 240 fps. LED fill lights were used to enhance the light intensity of the experimental environment. The fast camera was fixed by a triangular support frame to the diagonal front of the picking mechanism of the returning machine to increase the wide angle and thereby capture the sugarcane leaf-picking process. Sugarcane leaves obtained from the farm of Ningxi Teaching and Research Base of the South China Agricultural University in March 2021 were used. Sugarcane leaves of 1000 mm in length, naturally dried and with a moisture content of 4.4% were randomly selected for the experiment. The sugarcane leaf was placed in parallel on the flat ground in front of the picking mechanism, which is almost the same as the sugarcane leaf in the simulation model.

*4.2. Experimental Results and Analysis*

4.2.1. Analysis of Sugarcane Leaf Posture Changes

The quick camera recorded the process of the sugarcane leaf being picked up by the picking mechanism, as shown in Figure 16.

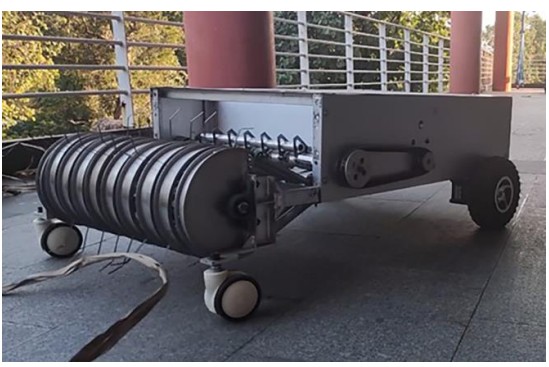

(**a**)

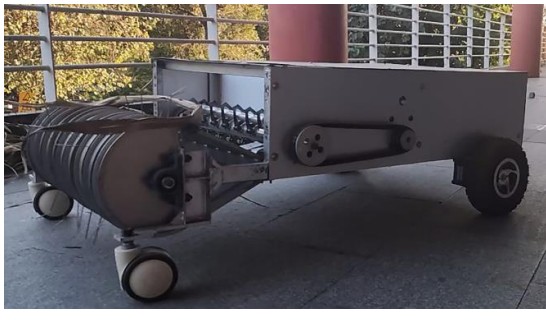

(**b**)

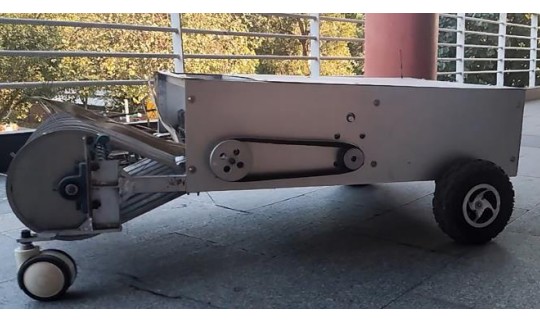

(**c**)

**Figure 16.** Quick camera experiment of sugarcane leaf-picking process. (**a**) represents the situation for the experiment at 0.2325 s, (**b**) represents the situation for the experiment at 1.0695 s, and (**c**) represents the situation for the experiment at 1.86 s.

It can be seen that at 0.2325 s, the middle section of the cane leaf had been picked up by the elastic teeth. Both sides of the sugarcane leaf were bent by the elastic teeth, and then hung down to touch the ground. The overall posture of the cane leaves is a "C" shape. At 1.0695 s, the sugarcane leaf was lifted above the guard by the elastic teeth and then pushed downwards by the elastic teeth. The bending of the sugarcane leaf on both sides had become greater. The overall posture of the sugarcane leaf was in the shape of a logarithmic curve. At 1.86 s, the two sides of sugarcane leaf hit the door frame. The middle of the sugarcane leaf stopped at the end of the guard plate. At this time, the whole posture of the sugarcane leaf is a "V" shape. The posture of sugarcane leaf in the experiment is in good agreement with that in the simulation.

### 4.2.2. Analysis of Changes in Sugarcane Leaf Bending Angle

The data on the change in the bending angle of the sugarcane leaf were extracted using a computer, and the bending angle was calculated using the same formula as in the simulation analysis. The experimental data of quick camera were compared with the data from the simulation. The comparison between the experimental value and simulation value of the sugarcane leaf bending angle is shown in Figure 17.

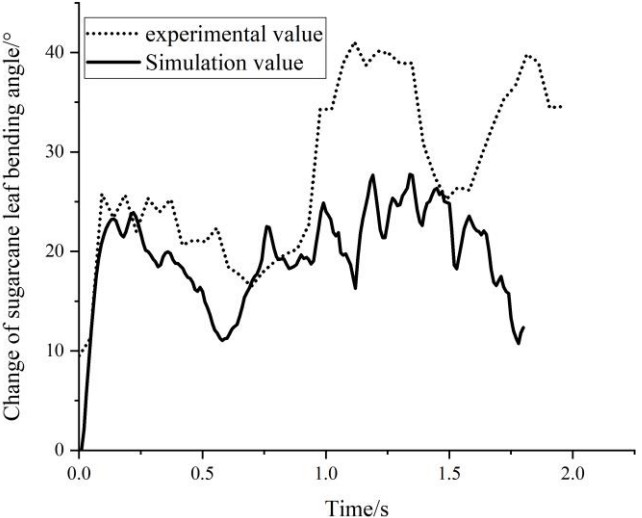

**Figure 17.** Comparison of experimental and simulated values of bending angle change.

Due to the characteristics of the leaf and the way it was stored, the initial bending angle was approximately 10° when the leaf first came into contact with the elastic teeth in the experiment. This was a bit different from the ideal arrangement of the cane leaves in the simulation, which appeared to be straight and flat. Firstly, the bending angle of the sugarcane leaf increased rapidly before 0.14 s in simulation. Secondly, the sugarcane leaf folding angle gradually decreased overall during the period of 0.14~0.58 s. Thirdly, in the period of 0.58~1.34 s, the folding angle generally increased in a fluctuating way, reaching a maximum value at 1.34 s. Finally, the folding angle of the sugarcane generally fluctuated in the range of 12° to 27°.

Initially, the bending angle of sugarcane leaf increased rapidly before 0.09 s in the experiment. After that, in the period of 0.09~0.70 s, the bending angle in the experiment generally decreased slowly. Then, in the period of 0.70~1.12 s, the bending angle in the experiment generally increased in a fluctuating way. Lastly, the bending angle generally fluctuated in the range of 25° to 40°. The overall trend in the bending angle of the sugarcane leaf was the same in the simulation and the experiment.

Both are made up of four stages, in which the bending angle firstly increased rapidly (stage 1), secondly decreased gradually (stage 2), thirdly increased in fluctuation (stage 3), and finally fluctuated within the range of 15 degrees' difference (stage 4). The correlation coefficient between the experimental and simulated bending angle change curves is 0.557.

The experimental value and simulation values of the average value of sugarcane leaf bending angle were 27 °and 19°, respectively, with a relative error of 29.6%. The moisture content of the sugarcane leaf will affect the posture of the sugarcane leaf. The reason for the large relative error is that the moisture content of the sugarcane leaf in the simulation and the moisture content of the actual sugarcane leaf cannot be guaranteed to be identical.

## 5. Conclusions

The picking contact mechanism of the electric sugarcane leaf cutting and returning machine is still unclear, and there are large gaps in the research on the interaction between the sugarcane leaf and the picking mechanism. It is necessary to analyze the picking process of the sugarcane leaf in detail. Therefore, in this paper, the finite element model of the picking system was established. The dynamic simulation of the picking system was carried out, and the quick camera experiment was conducted. The main conclusions are as follows:

(1) The working process of the elastic teeth roller picking mechanism was analyzed. The rotary position relationship of the roller plate corresponding to the posture change of the elastic teeth in the movement process was defined as the pick-up phase of the elastic teeth, which is expressed by the rotation angle of the roller plate. The changes in the movement of the elastic teeth during a cycle can be divided into four stages, namely the pick-up stage, the lift-up stage, the push-up stage and the retraction stage. Each of these four stages corresponds to the phase angle of the corresponding elastic teeth.

(2) A detailed analysis of sugarcane leaf postures, stresses, bending angles and stresses in the elastic teeth was carried out. The results showed that the posture changes of sugarcane leaves during picking were a "C", a logarithmic curve, a wavy shape and a "V" shape in turn. The maximum Von Mises stress of sugarcane blade and vein during picking was 22.8 MPa and 17.5 MPa, respectively. The sugarcane leaf showed minor breakage during the picking process, while the vein remained undamaged. The evaluation criterion of the bending angle was creatively put forward to measure the bending deformation of leaves. The overall bending angle of the sugarcane leaf first increased rapidly, then decreased gradually, then increased in a fluctuating way, and finally fluctuated within an interval of 15 degrees' difference. The maximum Von Mises stress of elastic teeth during picking was 900 MPa. The elastic teeth do not fail during the picking process, and the multi-segment linear plasticity model can be used to simulate the deformation of the elastic teeth.

(3) The results of the quick camera experiment were in good agreement with those of simulation analysis, which indicated that the simulation model is reliable and effective in analyzing the process of the picking up of sugarcane leaf by the picking mechanism. The change in the posture of the sugarcane leaf and the bending angle in the experiment were consistent with the simulation analysis. The trend for both the experimental and simulated sugarcane leaf bending angles was a four-stage process, with an overall rapid increase followed by a gradual decrease, then a fluctuating increase and finally, a fluctuating change within the interval of 15 degrees' difference. The trend of the bending angle was the same for both the experiment and simulation. The correlation coefficient between the experiment curve and the simulation curve was 0.557. The experimental and simulation values of the average sugarcane leaf bending angle were 27° and 19°, respectively. The relative error of the average bending angle was 29.6%. The reason for the large relative error is that the moisture content of the sugarcane leaf in the simulation and the moisture content of the actual sugarcane leaf cannot be guaranteed to be identical.

In this paper, the numerical simulation and experimental study of a forward speed and a roller plate speed were accomplished. In fact, different combinations of forward speed and drum speed have certain influence on the picking effect of the sugarcane leaf. When the speed of the roller plate is too high, the sugarcane leaf will be pushed away by the elastic teeth, resulting in a failure to pick up the sugarcane leaf. In the future, numerical simulation and experimental research on different combinations of forward speed and roller

plate speed will be carried out. The current numerical model has the shortcomings of long calculation time and some differences from the actual situation. In the future, the number of grids will be reduced appropriately, which will make the calculation time shorter and improve the calculation efficiency. What is more, the stress–strain curve of sugarcane leaf will be obtained through experiments, and then sugarcane leaf will be simulated through a more accurate material model to improve the accuracy of numerical simulation.

**Author Contributions:** Conceptualization, Z.Y. and Y.W.; methodology, Z.Y.; software, Z.Q.; validation, Z.Y., W.L. and G.R.; formal analysis, Z.Y.; investigation, W.L.; resources, Y.W. and Y.T.; data curation, Q.Z.; writing—original draft preparation, Z.Y.; writing—review and editing, Z.Y. and Y.W.; visualization, Y.T.; supervision, Y.W.; project administration, Y.W.; funding acquisition, Y.W. All authors have read and agreed to the published version of the manuscript.

**Funding:** The research was funded by the Natural Science Foundation Project of Guangdong Province, grant number 2021A1515010649.

**Institutional Review Board Statement:** Not applicable.

**Informed Consent Statement:** Not applicable.

**Data Availability Statement:** Not applicable.

**Acknowledgments:** The authors would like to thank all the reviewers and editors for their dedication, and the Guangdong Natural Science Foundation Management Committee for its support.

**Conflicts of Interest:** The authors declare no conflict of interest.

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
