# Peer review of "Dynamic Simulation Analysis of the Working Process of the Picking Mechanism of a Sugarcane Leaf Cutting and Returning Machine"

_applsci, doi:10.3390/app13031620_

Round 1

Reviewer 1 Report

Below are some highlights to make this more publication-ready:

1.     Some typo errors in the article should be corrected.

2.     It is recommended to add some suggestions for future works in this area to improve the conclusion.

3.     The Abstract should contain answers to the following questions: What problem was studied, and why is it important? What is the novelty of the work?

4.     The "Introduction" section should be impressive and informative. Some relative papers are recommended to improve this section. https://doi.org/10.1016/j.compag.2021.106538

5.     The authors should try to give advantages of using their method compared to others.

6.     The conclusion section can be improved.

7.     In conclusion and also in the abstract, authors must present their main outcomes in quantitative outcomes.

8.     I recommend expanding: Introduction, Conclusions and the Results sections. The aim should be to give a broader view of the literature on the topic and the current state-of-the-art. clarify and discuss the novelty and the significance of the results obtained here, and compare them with those available in the literature.

9.     All equations are to be followed by either Comma or full stop.

10.  There are concerns about the grammar, usage, and overall readability of the manuscript. It is not possible to mention all the grammatical errors here. The authors are recommended to carefully read the manuscript again and remove all mistakes.

11.  Present a more focused survey on the specified topic. Also, at the end of the Introduction, clarify the novelty and gaps to be filled in the literature by the present attempt.

12.  Maintain uniform size symbols and equations through manuscript.

Reviewer 2 Report

In this paper, the authors studied by a numerical and experimental investigation the phenomenon of sugarcane leaf picking. Indeed, they modeled the mechanical system used for this process under the ANSYS/ LS-DYNA interface in order to analyze the posture of the sugarcane leaf during the picking process and to evaluate the behavior of some machine components during this process. At the end of their manuscript, the authors made a comparison between numerical and experimental results in terms of changes in the bending angle of the sugarcane leaf.

In the reviewer opinion, the paper can be recommended for publication in Applied science journal after addressing the following comments:

- Section 2. It is recommended that dimensional specifications of key components of the mechanism used for the sugarcane picking process be added.

- The elastic teeth geometric parameters should be added

- Line 153 the authors specified that they used isotropic and kinematic hardening plasticity, however they should specify in the material characteristics the coefficients used for this hardening model and what precise model was used (e.g. Hill`48 hardening model).

- What is the material type of the elastic teeth?

- What is the behavioral model of the material used for these elastic teeth?

- The measured stress values in Figure 5 are not readable.

- Line 291. The authors should state the reason why the maximum value of stress on the leaf is measured at this particular time t=1.53 s. As well as the value measured at this time for the elastic teeth from figure 7.

- The values of the effective stress plotted in figure 8 are not readable.

- The evolution of the bending angle of the sugarcane leaf during the picking process plotted in figure 10 seems to be an arbitrary curve and will not be the same curve for another leaf which means that this curve has no meaning.

Figure 14. Looking at this figure, there is a remarkable difference between the numerical result and the experimental result that needs to be explained. The numerical model should be improved to be more realistic.

- The conclusion ``Comparing the experiment with the simulation, the research showed that the simulation data was consistent with the experimental data, and the finite element model was in general agreement with the actual situation``. Seems to be not correct when looking at this figure (14).

Round 2

Reviewer 2 Report

The present form can be accepted for publication